# Individuals with autism spectrum disorder have altered visual encoding capacity

**Jean-Paul Noel**[1‡], **Ling-Qi Zhang**[2‡], **Alan A. Stocker**[2,3‡], **Dora E. Angelaki**[1‡*]

**1** Center for Neural Science, New York University, New York City, New York, United States of America,
**2** Department of Psychology, University of Pennsylvania, Philadelphia, Pennsylvania, United States of America, **3** Department of Electrical and Systems Engineering, University of Pennsylvania, Philadelphia, Pennsylvania, United States of America

‡ JPN and LQZ share first authorship on this work. AAS and DEA are joint senior authors on this work.
* da93@nyu.edu

**Data Availability Statement:** The data underlying each figure can be found in supplementary files. Additional data and code for our analysis can be

## Abstract

Perceptual anomalies in individuals with autism spectrum disorder (ASD) have been attributed to an imbalance in weighting incoming sensory evidence with prior knowledge when interpreting sensory information. Here, we show that sensory encoding and how it adapts to changing stimulus statistics during feedback also characteristically differs between neurotypical and ASD groups. In a visual orientation estimation task, we extracted the accuracy of sensory encoding from psychophysical data by using an information theoretic measure. Initially, sensory representations in both groups reflected the statistics of visual orientations in natural scenes, but encoding capacity was overall lower in the ASD group. Exposure to an artificial (i.e., uniform) distribution of visual orientations coupled with performance feedback altered the sensory representations of the neurotypical group toward the novel experimental statistics, while also increasing their total encoding capacity. In contrast, neither total encoding capacity nor its allocation significantly changed in the ASD group. Across both groups, the degree of adaptation was correlated with participants' initial encoding capacity. These findings highlight substantial deficits in sensory encoding—independent from and potentially in addition to deficits in decoding—in individuals with ASD.

## Introduction

Autism spectrum disorder (ASD) is a neurodevelopmental condition with high prevalence [1] which ubiquitously afflicting brain function, from perception to (social) cognition. Recent reports have attempted to formalize the functional implications of the disorder within the language of statistical inference, proposing that across domains ASD is largely characterized by an imbalance in weighting incoming sensory evidence with prior knowledge about the world [2–9].

This formulation places the study of ASD within the encoding–decoding framework of perception (**Fig 1A**). Namely, sensory stimuli, $\theta$, are first encoded by a noisy neural measure—i.e., sensory information is mapped onto bandwidth-limited neural responses. Then, for example, within the framework of statistical (Bayesian) inference, these measurements, $m$, inform an

found at: https://github.com/cpc-lab-stocker/ASD_Encoding_2020.

**Funding:** This work was supported by the Simons Foundation, via a SFARI Grant (#396921) to D.E.A. The funders had no role in study design, data collection and analysis, decision to publish, or preparation of the manuscript.

**Competing interests:** The authors have declared that no competing interests exist

**Abbreviations:** AQ, Autism Spectrum Quotient; ASD, autism spectrum disorder; FI, Fisher information; MLE, maximum likelihood estimation; RMSE, root-mean-square error; RT, response time; SCQ, Social Communication Questionnaire.

observer about the identity or magnitude of a stimulus variable (likelihood). Combined with prior expectations about the environment, $p(\theta)$, this results in a posterior belief $p(\theta|m)$. An estimate of the stimulus variable $\hat{\theta}(m)$ is generated based on the posterior and the utility function participants employ for the task. Together, these components and their appropriate combination (i.e., combination of a likelihood with a prior resulting in a posterior that is used in accordance with a utility function to generate responses) represent the "decoding" of sensory representation. Anomalies in this decoding process have been the main focus of recent computational studies of ASD (e.g., imbalance in the weighting of likelihood versus priors; see citations above), with researchers recently also emphasizing a slow or inflexible updating of priors in ASD [10,11].

The optimal combination of likelihoods with priors is, however, only one of many potential decoding mechanisms. Other commonly studied decoders in perceptual science include maximum likelihood estimation (MLE; [12]), linear decoders (e.g., [13]), and in more recent years, decoders based on neural networks [14]. Given the array of potential decoders the brain may employ and the various ways in which these may malfunction in ASD [8], here we were instead interested in identifying potential deficits in the sensory encoding, and the updating of this encoding with sensory exposure and feedback, irrespective of differences in decoding. This exclusive emphasis on sensory encoding is seldom, if ever, explored within the ASD literature and does not detract from reported anomalies in sensory decoding. Rather, it provides the opportunity to cleanly disentangle and localize deficits in encoding from deficits in decoding.

As experimenters, however, we can only measure participants' responses as reported by verbal or motor output. These reports are influenced by both encoding and decoding processes (including participants' prior belief and utility function, or "response model"), thus rendering the characterization of a singular process (i.e., encoding or decoding) difficult. To address this challenge, and to specifically test the hypothesis that sensory encoding and its updating are aberrant in ASD, we leverage a principled framework for data analysis emphasizing sensory encoding independently from decoding (**Fig 1B and 1C**). Specifically, we compute Fisher information (FI), a measure of how much information participants' internal representations carry about the physical stimulus presented, by using a lawful relationship between estimation bias, variance, and FI, known as the Cramer–Rao lower bound (**Eq 1**; [15,16]):

$$\sqrt{I_F(\theta)} \geq \frac{[1 + b'(\theta)]}{\sigma(\theta)}. \tag{Eq 1}$$

In other words, by measuring bias (b) and variance ($\sigma^2$) in estimates ($\hat{\theta}$) of $\theta$, we can determine a lower bound on FI, i.e., the accuracy with which the stimulus variable is represented. Equating this bound with a direct index of sensory encoding (i.e., in effect replacing the greater than or equal sign with an equality in **Eq 1**) requires a single assumption—that the translation from sensory representations to behavioral outputs was uncorrupted by stimulus-dependent noise. Note that this assumption is (often implicitly) made by virtually all decoders in perceptual science (e.g., Bayesian decoders, MLE), and neural decoders (e.g., linear classifiers, population vector, etc.). The exact form of the decoder (i.e., idiosyncrasies in the mapping from sensory estimates to motor outputs) does not matter beyond this single assumption, given that any decoder will trade off bias and variance. We further motivate and demonstrate the validity of our approach via both a numerical simulation (**Fig 1**) and an analytical derivation (see **S1 Text**) showing that: (1) the same encoding process (**Fig 1D**) can produce very distinct patterns of estimation bias and variance given different decoders (**Fig 1E**), yet critically; (2) with the Cramer–Rao lower bound, we are able to precisely recover the encoding scheme in terms of FI, regardless of the differences in decoding (**Fig 1F**).

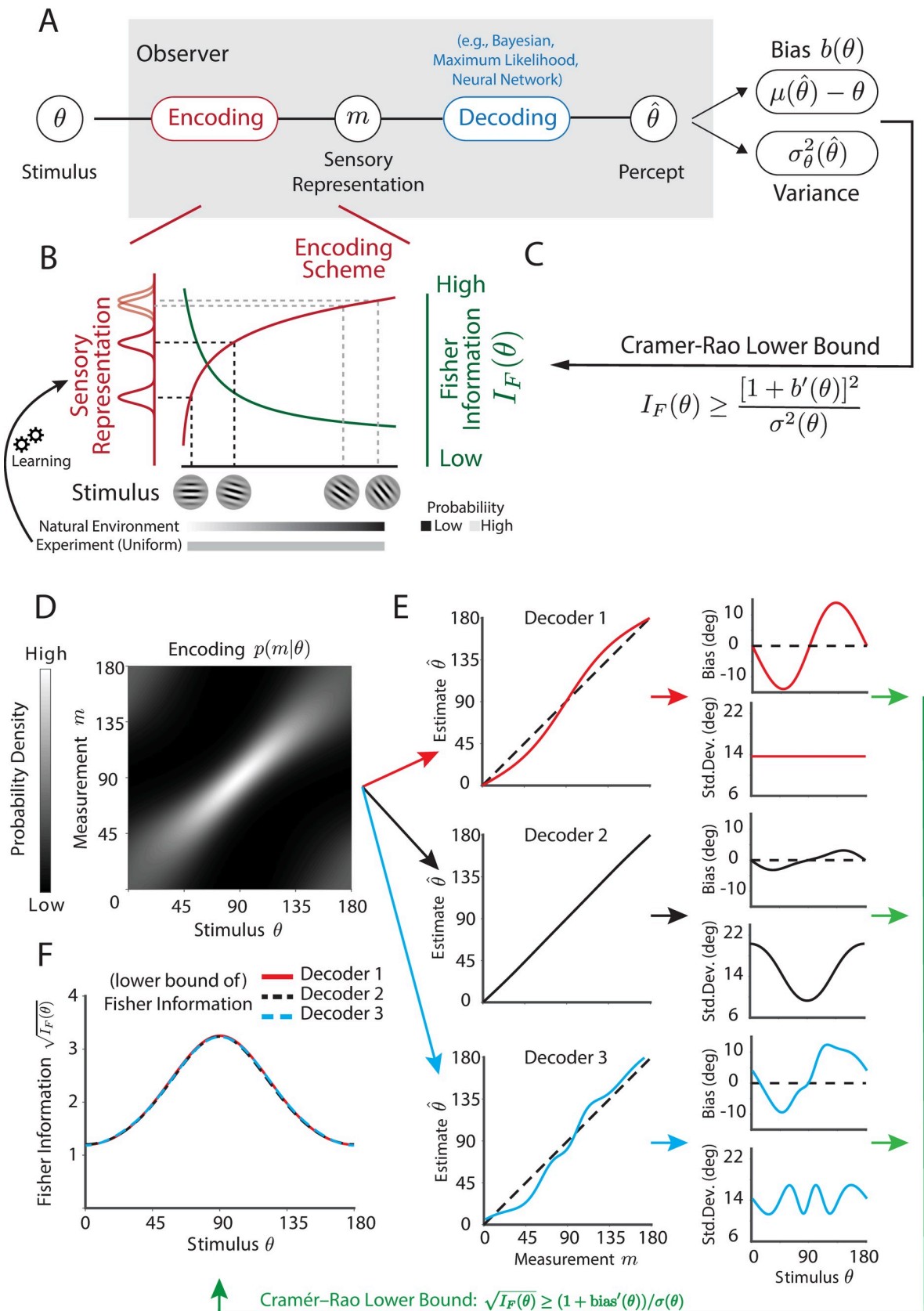

**Fig 1. Theoretical and conceptual framework.** (**A**) Perception can be described as an encoding–decoding process. Stimulus is first encoded in the noisy and resource limited sensory representation $m$. An estimate $\hat{\theta}$ is generated given $m$ based on the decoding process. As an experimenter, we can only measure overt responses; characterized as a participant's bias ($\mu(\hat{\theta}) - \theta$; difference between the average estimate and real stimulus value) and variance $\sigma^2(\theta)$ in their responses. (**B**) The fidelity of encoding in neurotypical participants is generally anisotropic due to uneven allocation of sensory resources determined by environmental statistics. For example, as shown here, the nonlinear transformation between stimulus values and a neural space (solid red line) with homogenous noise results in higher uncertainty for orientations around the oblique than those around the cardinal. This can be characterized by a nonuniform profile of Fisher information $I_F(\theta)$ (solid green line). In our experiment, we imposed an artificial, uniform distribution of orientations, different from that of natural environment, which allows us to study whether and how both groups update their encoding under changing stimulus statistics. (**C**) Cramer–Rao lower bound specifies the lawful relationship between bias $b(\theta)$, variance $\sigma^2(\theta)$, and encoding accuracy, i.e., Fisher information $I_F(\theta)$, regardless of the decoding scheme. This allows us to directly characterize encoding, while remaining agnostic about the details of the decoding process (also see S1 Text). (**D**) To demonstrate validity of our approach, we simulate an observer with anisotropic encoding process, $p(m|\theta)$, with a peak FI at 90˚. (**E**) As an example, we construct 3 arbitrary decoders (red, black, and blue), yielding very distinct pattern of estimation biases and variances, yet they all attain the Cramer–Rao lower bound. (**F**) Applying the inequality, we estimate the (lower bound of) FI, which appropriately recovers the identical, true underlying pattern of FI in the encoding regardless of idiosyncrasies in the decoding process. Raw data and code underlying this figure can be found at S1 Code, and numerical values that make up this figure can be found at S1 Data. FI, Fisher information.

To identify anomalies in the capacity, and the updating of sensory encoding in ASD, we took advantage of the well-known "oblique effect" [17,18]. That is, humans demonstrate greater sensitivity and repulsive (i.e., away from) biases in perceiving gratings and motion around cardinal compared to oblique orientations [19,20]. This effect is thought to reflect the fine-tuning of sensory encoding to environmental statistics—with vertical and horizontal orientations being most common in nature [21,22]. Recent studies [16,23] further highlight that it is exactly this anisotropic property of sensory encoding that leads to the observed repulsive bias. In other words, the oblique effect is thought to reflect a property of sensory encoding, and not decoding. A standard Bayesian decoder with homogenous encoding cannot account for the oblique effect, given the bias is away from and not toward the prior [23]. Lastly, early studies demonstrated that orientation discrimination is readily impacted by perceptual learning, adaptation, and feedback (e.g., [24–26]) on a relatively short timescale. In turn, orientation estimation is an appropriate paradigm to study the updating and flexibility of the encoding process.

First, we had neurotypical and ASD individuals estimate orientations in the absence of feedback, in order to index their baseline accuracy in encoding individual stimulus orientations and overall encoding capacity. Then, to parallel observations emphasizing the slow updating of Bayesian priors in ASD [10,11] and establish whether this phenotype would extend to sensory encoding, we presented participants with an experimental distribution of orientations (i.e., uniform) that differs from the natural distribution (i.e., peaking at cardinal orientations) and provided participants with performance feedback. Feedback was given in order to accelerate any putative reallocation of encoding resources due to the change in orientation distribution.

## Results

### Heightened variability and reduced learning in autism

We asked participants to perform a visual orientation estimation task, first without feedback, and then in the presence of trial-by-trial feedback in an attempt to induce learning and updating of sensory encoding. Groups of ASD ($n = 17$) and neurotypical ($n = 25$) individuals matched via button press a briefly (120 ms) presented visual target of random orientation. Target orientations were drawn from a uniform distribution. Initially, no feedback was presented (woFB block), but in the second and third block of trials (wFB1 and wFB2; 200 trials/block), feedback was presented by overlaying the participant's response and the target (**Fig 2A**, see Materials and methods for details). As shown for an example neurotypical (**Fig 2B**) and ASD

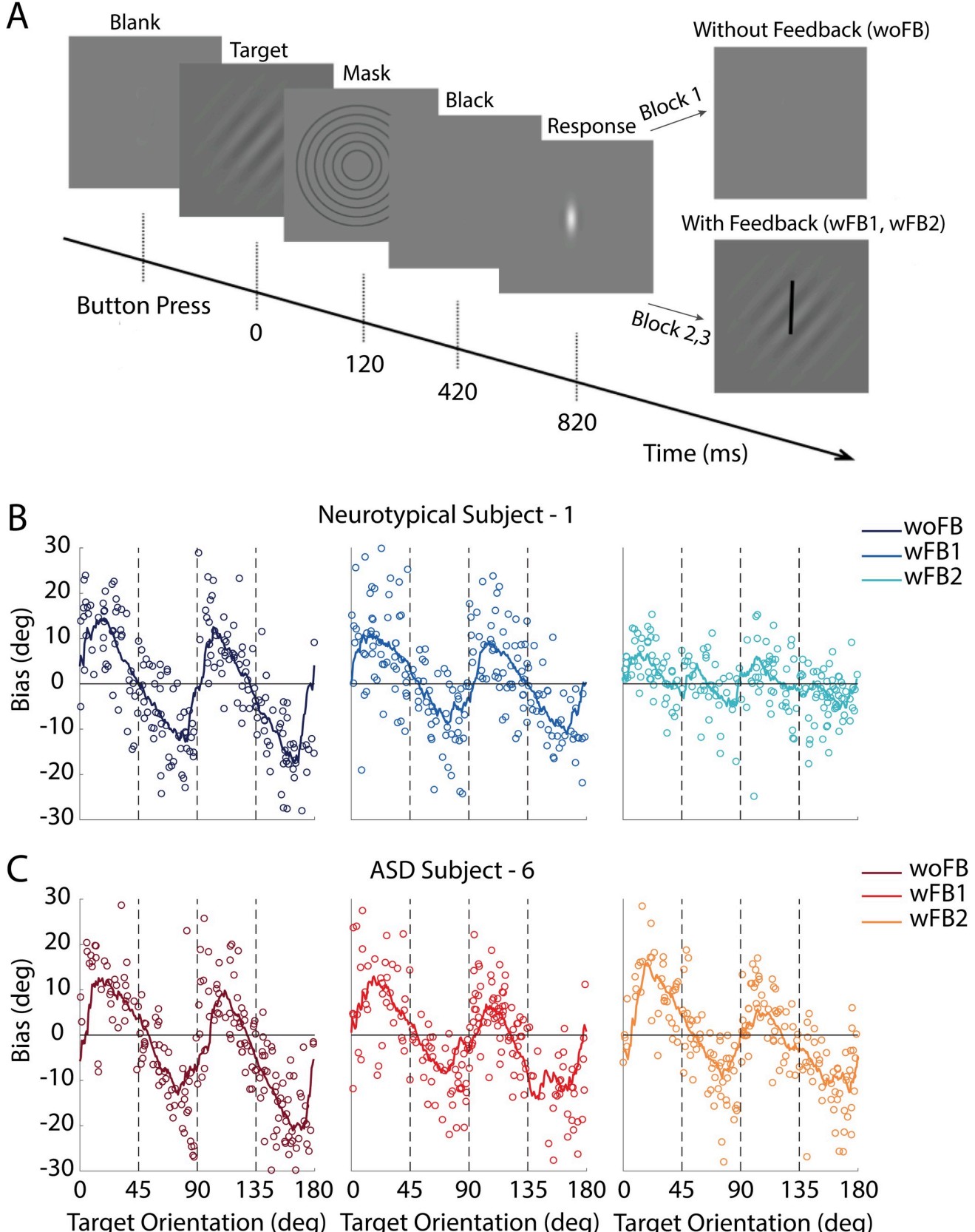

**Fig 2. Experimental protocol and individual participant performance.** (**A**) A target orientation (Gabor) is briefly presented, and participants report their percept by orienting a line indicator (white) via left or right button press. No feedback is given on the first block of trials but is in subsequent blocks by overlaying the target orientation and the participant's response. (**B**) Target orientations (x-axis) are drawn from a uniform distribution (individual dots are single trials). Y-axis indicates the bias for an example neurotypical participant, and lines are the running average within a sliding window of 18°. Different columns (and the color gradient) respectively show performance on the block without feedback (woFB; leftmost), on the first block with feedback (wFB1; center), and the second block with feedback (wFB2; rightmost). (**C**) follows the format in (**B**) while depicting the performance of an example ASD participant. Raw data and code underlying this figure can be found at **S1 Code**, and numerical values that make up this figure can be found at **S2 Data**. ASD, autism spectrum disorder.

(**Fig 2C**) individual, orientation perception was biased away from cardinal orientations in both groups—namely, both groups demonstrated an "oblique effect" [17,18]. With feedback, the bias seemingly dissipated in the neurotypical participant but not the ASD individual (**Fig 2B and 2C** scatter plot are individual responses, curve is a running average within a sliding 18° window, and vertical dashed lines are orientations at 45°, 90°, and 135°. See **S1 and S2 Figs** for similar plots for all individual participants).

These basic observations are also evident in group averages of neurotypical and ASD individuals (**Fig 3**). When presenting targets between 0° (horizontal) and 45°, the bias was on average positive (e.g., neurotypical group: $6.0° \pm 0.26°$ (SE), $p < 10^{-3}$), suggesting that horizontal

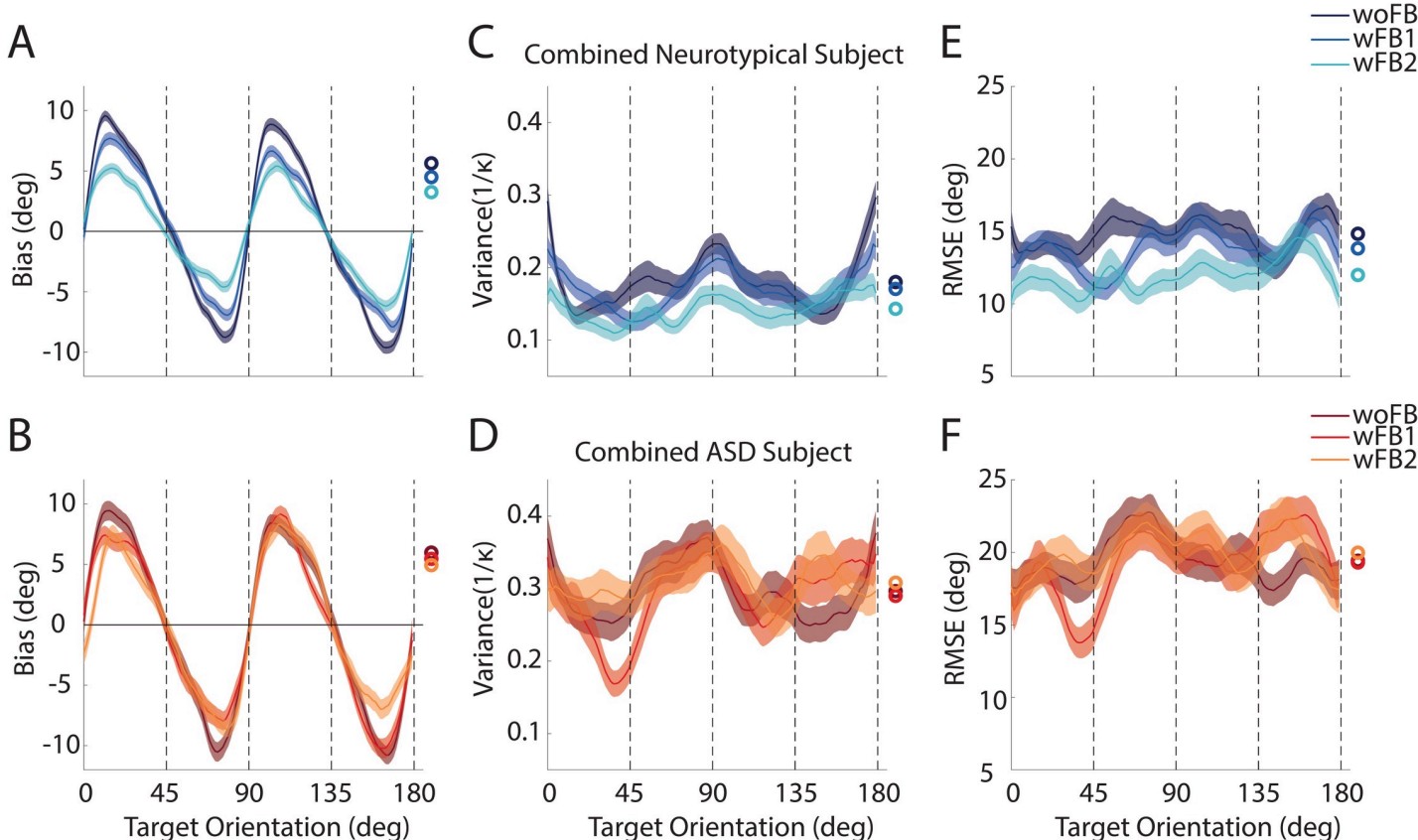

**Fig 3. Orientation perception in combined neurotypical and ASD participant.** Bias (y-axis) as a function of target orientation (x-axis, cardinal and oblique orientation indicated by dashed lines) and feedback block in neurotypical (**A**) and ASD (**B**) participants. Variance (1/κ, y-axis, see Materials and methods) as a function of target orientation and feedback block in neurotypical (**C**) and ASD (**D**) participants. RMSE (y-axis) as a function of target orientation and feedback block in neurotypical (**E**) and ASD (**F**) participants. Smoothing using a sliding window of 18° has been applied for visualization purpose. Error bars are ± SEM across 5,000 bootstrap runs. Raw data and code underlying this figure can be found at **S1 Code**, and numerical values that make up this figure can be found at **S3 Data**. ASD, autism spectrum disorder; RMSE, root-mean-square error.

gratings were perceived closer to the oblique 45°. Contrary, when presenting targets between 45° and 90° (vertical), the bias was negative (e.g., neurotypical group: $-5.14° \pm 0.26°$, $p < 10^{-3}$), suggesting that vertical gratings were again perceived closer to the oblique 45° (**Fig 3A and 3B**). Before feedback, there was no statistical difference in the overall magnitude (i.e., absolute value) of bias between neurotypical and ASD groups (neurotypical: $5.63° \pm 0.12°$; ASD: $5.99° \pm 0.22°$, $\Delta = 0.35° \pm 0.25°$, $p = 0.154$). On the other hand, when provided with feedback, orientation perception bias was reduced in the neurotypical group but much less so in the ASD group (reduction in average magnitude of bias between woFB and wFB2, neurotypical: $\Delta = 2.38° \pm 0.17°$, $p < 10^{-3}$; ASD: $\Delta = 1.04° \pm 0.31°$, $p < 10^{-3}$; $\Delta$Neurotypical—$\Delta$ASD = $1.34° \pm 0.35°$, $p < 10^{-3}$; **Fig 3A and 3B,** feedback conditions shown by a color gradient).

Regarding the variability of orientation perception, we found that at baseline (i.e., before feedback), the ASD group had larger variance in their estimates than the neurotypical group (neurotypical: $0.180 \pm 0.006$; ASD: $0.297 \pm 0.010$; $\Delta = 0.117 \pm 0.11$, $p < 10^{-3}$; **Fig 3C and 3D**). This is in line with a growing literature suggesting heightened variability in ASD and the "neural unreliability thesis" of ASD [27–31]. Additionally, while variance was reduced with feedback in the neurotypical group, especially in the latter feedback block, this was not evident in the ASD cohort (reduction in overall variance between woFB and wFB2, neurotypical: $\Delta = 0.037 \pm 0.008$, $p < 10^{-3}$; ASD: $\Delta = -0.012 \pm 0.015$, $p = 0.414$).

As expected, given these differences in bias and variability, the neurotypical group overall had better performance as measured by root-mean-square error (RMSE). This was true in the initial block of the experiment (neurotypical: $14.82° \pm 0.35°$, $p < 10^{-3}$; ASD: $19.42° \pm 0.47°$, $p < 10^{-3}$) and was exacerbated with feedback (**Fig 3E and 3F**). Namely, performance of the neurotypical, but not the ASD, group increased with feedback (reduction in overall RMSE between woFB and wFB2, neurotypical: $\Delta = 2.83° \pm 0.48°$, $p < 10^{-3}$, ASD: $\Delta = -0.55° \pm 0.71°$, $p = 0.78$).

## Reduced capacity and aberrant allocation of encoding resources in autism

Beyond characterizing the raw task performance of neurotypical and autistic individuals, our main goal here is to detail their encoding capacity, the allocation of these resources, and flexibility with which encoding resources are reallocated, given the artificial stimulus distribution presented in the experiment. Using the Cramer–Rao bound (**Eq 1**) we found that in both neurotypical and ASD individuals, FI peaked at cardinal orientations (**Fig 4A**). That is, initially the shape of FI (i.e., normalized $\sqrt{I_F(\theta)}$, see Materials and methods, **Eq 3**) in both groups qualitatively matched the previously measured distribution of orientations in natural images [21,22]. Additionally, since FI directly determines discrimination thresholds [16,32], the observed shape of FI also corroborates a heightened sensitivity to cardinal orientations [18] in both neurotypical and individuals with ASD. On the other hand, both total FI (i.e., $\int_\theta \sqrt{I_F(\theta)} \, d\theta$) during the very first block of trials and how its magnitude changed over the course of the experiment differed for ASD and neurotypical groups. Already during the woFB block, total FI was significantly lower in ASD than neurotypical individuals (woFB, neurotypical: $14.67 \pm 0.24$, ASD: $11.28 \pm 0.19$, $\Delta = 3.39 \pm 0.43$, $p < 10^{-3}$). Further, total FI increased from woFB to wFB2 for the neurotypical group ($\Delta = 1.96 \pm 0.38$, $p < 10^{-3}$) but did not in the ASD group ($\Delta = -0.060 \pm 0.27$, $p = 0.83$, **Fig 4B**).

The above analyses suggest that similarly to their neurotypical counterparts, individuals with ASD have placed the bulk of their encoding resources in line with the distribution of orientations in the natural environment (i.e., at cardinal orientations). More broadly, this observation supports the efficient coding hypothesis [33], whereby encoding resources are characterized by FI and directly follow the shape of the stimulus distribution (see Materials and methods, **Eq 2**). In turn, to quantitatively assess encoding capacity and distribution of

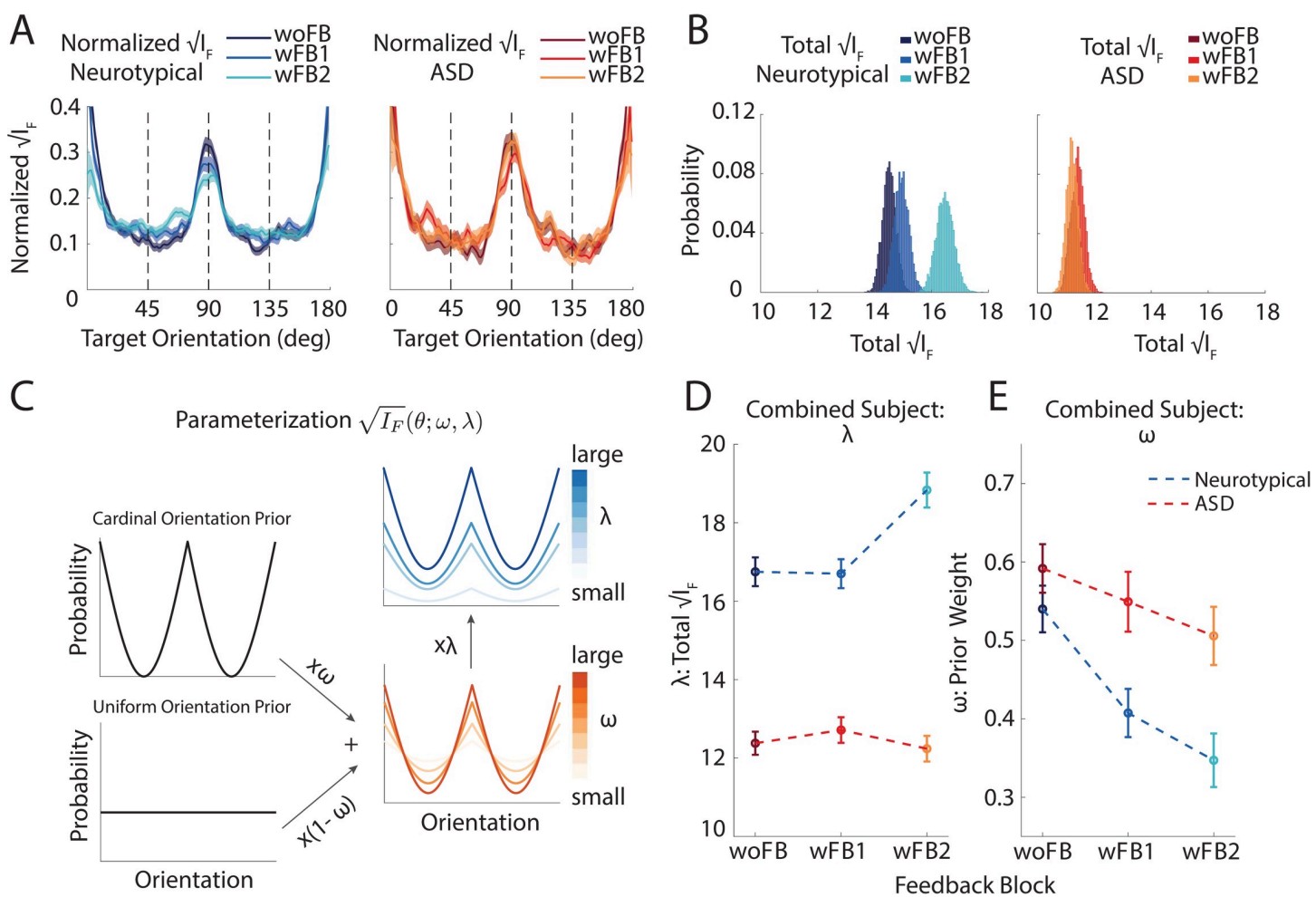

**Fig 4. Quantification and parametrization of FI in neurotypical and ASD individuals. (A)** FI peaked at cardinal orientations for both neurotypical (blue) and ASD (red) individuals, similar to the natural scene statistics of orientations. Further, visual inspection suggests a flattening of this function with feedback in neurotypical participants (from dark to light blue) but not ASD participants (see panels **C** and **E** for quantification). Smoothing using a sliding window of 18˚ has been applied for visualization. **(B)** The total amount of FI was larger in neurotypical individuals than ASD at the outset and increased over the course of the experiment in neurotypical (blue color gradient) but not ASD (red color gradient) individuals. **(C)** The FI pattern as a function of orientation was quantified by 2 parameters, $\omega$, which mixes a cardinal orientation prior with a uniform distribution as the normalized (square root of) FI, and $\lambda$, scaling total FI. **(D)** $\lambda$ and **(E)** $\omega$ as a function of group (blue = neurotypical control, red = ASD) and block. Error bars are ± SEM across 5,000 bootstrap runs. Raw data and code underlying this figure can be found at **S1 Code**, and numerical values that make up this figure can be found at **S4 Data**. ASD, autism spectrum disorder; FI, Fisher information.

resources in both groups throughout the experiment, we parametrized the shape of FI as a weighted mixture of a "cardinal orientation distribution" and a uniform orientation distribution (with weights $\omega$ and $1-\omega$, respectively; **Fig 4C**; also see Materials and methods, **Eq 4**). A $\omega$ value of 0.5 approximates the previously measured prior distribution of orientation in natural images [21,22]. Presumably, allocation of encoding resources starts closer to the natural stimulus distribution for orientations ($\omega{\sim}0.5$) and is then gradually molded to become more uniform (i.e., smaller $\omega$). Thus, by parametrizing normalized $\sqrt{I_F(\theta)}$ as a weighted sum of the cardinal orientation prior and the distribution of orientations presented during the experiment (i.e., uniform), we can quantify if and how feedback changed our participants' allocation of encoding resources along a range, from the natural to the experimental stimulus distribution. Further, the overall magnitude of FI is determined by a scaling parameter, $\lambda$ (**Fig 4C, Eq 4** in Materials and methods). We fit the bias predicted by the parameterized FI to the measured

bias given the measured variance (see Materials and methods, **Eqs 5 and 6,** and S1 and **S2 Figs** for the biases and fits of individual participants; see **S3** and **S4 Figs** for parameters extracted for individual participants). Average $R^2$ for the neurotypical group: 0.770 ± 0.025; average $R^2$ for ASD group: 0.774 ± 0.034.

Examination of the fitted $\lambda$ values confirmed that at the outset of the experiment, the ASD group had a lower total amount of FI compared to the neurotypical group (neurotypical versus ASD at woFB; $\Delta$ = 4.352 ± 0.473, $p < 10^{-3}$; **Fig 4D**). This effect is unlikely due to differences in task strategies, such as speed–accuracy trade-offs, given that the distribution of response times (RTs) was qualitatively similar across both groups, and if anything, slower in ASD, making this latter group deficitary in both speed and accuracy (**S5 Fig**). Over the course of the experiment, $\lambda$ increased further for the neurotypical group (woFB versus wFB2 in neurotypical individuals: $\Delta$ = 2.094 ± 0.580, $p < 10^{-3}$; **Fig 4D**)**,** but this was not the case for the ASD group ($\Delta$ = −-0.150 ± 0.441, $p$ = 0.732, **Fig 4D**). These results suggest a lower overall encoding capacity in ASD relative to their neurotypical counterparts and further a capacity that does not improve over the course of repeated feedback. On the other hand, we observed a significant decrease in median RT between woFB and wFB2 for both the neurotypical ($\Delta$RT = −0.61 ± 0.069 sec, $p < 10^{-3}$) and ASD group ($\Delta$RT = −0.75 ± 0.114 sec, $p < 10^{-3}$; see **S5 Fig**), again suggesting that differences in task strategy or putative differences in working memory are unlikely to account for the differing accuracy and precision in visual orientation estimation across ASD and neurotypical individuals.

Most interestingly, regarding participants' allocation of encoding resources, or their prior distribution of orientation as derived from a parametrized normalized $\sqrt{I_F(\theta)}$, both groups started with a similar $\omega$ parameter before feedback ($\Delta$ = 0.052 ± 0.043, $p$ = 0.230, **Fig 4E**). That is, these groups allocated resources similarly in the natural environment where horizontal and vertical edges are most common. When provided with feedback (woFB versus wFB2), the $\omega$ parameter decreased for the neurotypical group (change in $\omega$: $\Delta$ = 0.191 ± 0.046, $p < 10^{-3}$, **Fig 4E**, blue) but much less so for the ASD group (change in $\omega$: $\Delta$ = 0.086 ± 0.049, $p$ = 0.080). Again, under the efficient coding framework, this change in the $\omega$ parameter can be viewed as participants of the neurotypical group reallocating their encoding resources, from following the natural distribution of orientations to now incorporating the uniform stimulus distribution imposed by the experiment. The ASD group, instead, did not readily update their allocation of encoding resources (see **S3** and **S4 Figs** for individual participant data).

## Reduced initial encoding capacity is correlated with inflexibility in updating resource allocation

We further questioned whether the herein described anomalies in sensory encoding present in ASD may relate to their purported inflexibility in updating priors [10,11]. Indeed, recent theoretical studies [34,35] highlight an inherent relation between encoding resources and the degree to which one can adapt to changing environments. Intuitively, if resources are allocated primarily to represent statistically likely events, then fewer can be devoted to distinguishing between statistically unlikely alternatives—and it is precisely these latter ones that are most informative regarding a potential change in the environment [34,35]. To examine this hypothesis, we correlated participants' encoding capacity ($\lambda$) before feedback with the shape of their prior after feedback ($\omega$ at wFB2). We found a strong correlation between these variables ($R^2$ = 0.407, $p < 10^{-3}$, **Fig 5A**). The correlation holds separately for the neurotypical and ASD groups (**Fig 5B**). Thus, the inflexibility in resource reallocation observed in individuals with ASD may emanate from their limited encoding resources.

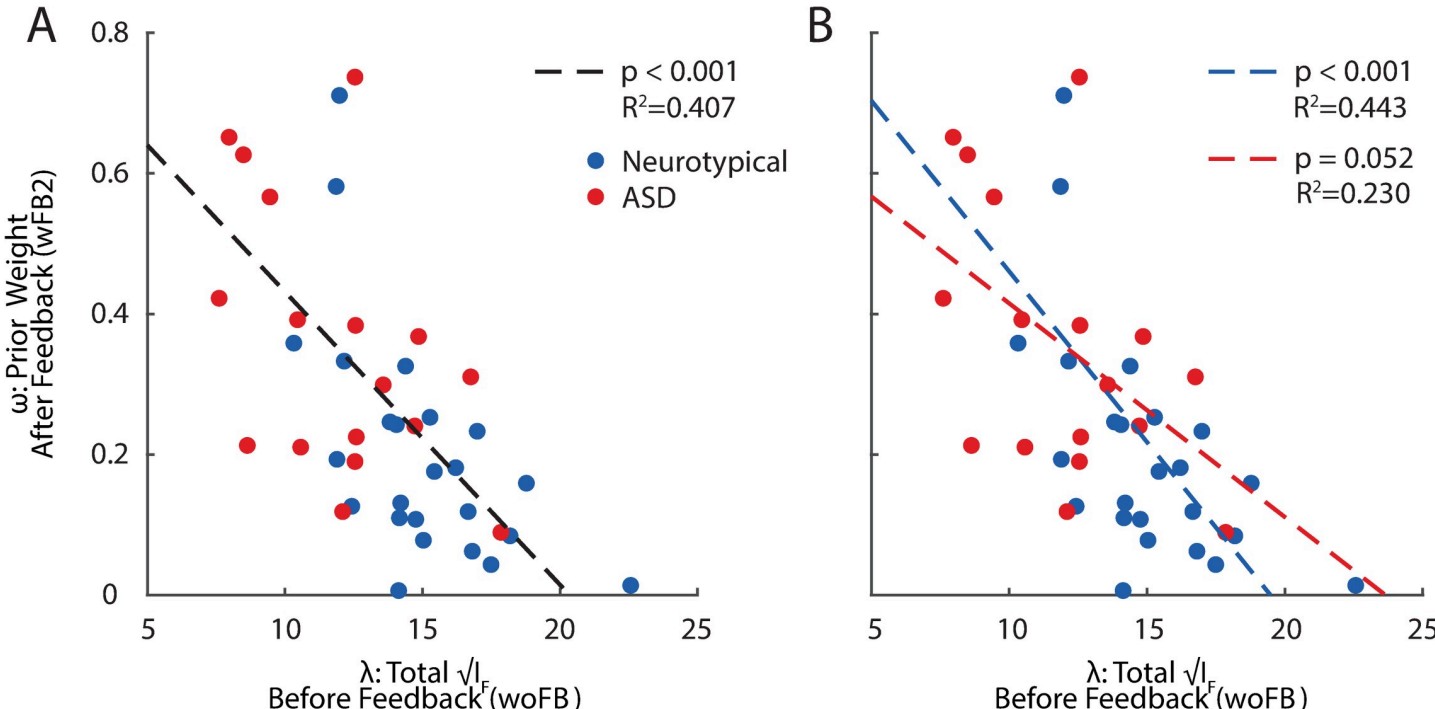

**Fig 5. Correlation between sensory encoding capacity and flexibility in encoding resources allocation.** (A) The flatness of FI after feedback (y-axis, indicating incorporation of stimulus statistics from the experiment) correlates strongly with total encoding resources before feedback (x-axis). The blue dots indicate individual neurotypical participants, and red dots indicate individual ASD participants. (B) The same correlation holds within both the neurotypical and ASD groups. Raw data and code underlying this figure can be found at **S1 Code**, and numerical values that make up this figure can be found at **S5 Data**. ASD, autism spectrum disorder; FI, Fisher information.

## Discussion

By tracking perceptual estimates while (i) exposing observers to an artificial stimulus distribution (i.e., uniformly distributed orientations) and presenting them with performance feedback, (ii) directly extracting FI of sensory encoding from psychophysical data, and (iii) parameterizing the shape of FI as a mixture of the cardinal and uniform distributions of orientation stimulus, we have estimated the total sensory encoding capacity and the change in its allocation with feedback in individuals with ASD. This combined approach resulted in a number of empirical observations. First, the initial pattern of FI as a function of orientation strongly resembled the known natural distribution of orientations in the natural environment [21,22] in both ASD and neurotypical individuals. Second, already in the first block of trials, the total encoding resources for the representation of gratings in ASD is about 2/3 that of their neurotypical counterparts. Third, exposure to an experimental distribution of orientations, coupled with feedback, resulted in the appropriate reallocation and an overall increase in encoding resources of neurotypical individuals. This was not the case in individuals with ASD. Lastly, initial encoding capacity correlated with the degree of adaptation from the natural to experimental allocation of encoding resources.

The observation that encoding capacity differs from the outset between ASD and neurotypical individuals is in line with "neural unreliability thesis" [27] and a number of recent reports that have suggested heightened variability in ASD, be it in relatively low-level visual psychophysics [36,37], more naturalistic and continuous tasks [31], neurovascular [28,29] or physiological responses [27]. Particularly noteworthy and relevant to the current report is a study by

Park and colleagues [37], where researchers had neurotypical and ASD individuals discriminate visual orientations. Then, via psychophysical modeling [38,39], they were able to ascertain that individuals with ASD had increased internal noise and a reduced capacity for external noise filtering than neurotypical counterparts. These results are also in line with accounts suggesting a reduced use of Bayesian priors [2] or an overweighting of sensory likelihoods versus expectations in ASD [5]. Namely, these expectations in essence serve as low-pass filters reducing trial-to-trial variability, and thus a decreased use of Bayesian priors could engender greater variability.

The second major observation is that the reallocation of sensory encoding resources is inflexible in ASD relative to neurotypical individuals. Given that within the efficient coding hypothesis the allocation of encoding resources may directly reflect participants' prior in the Bayesian framework [23], these results are similarly consistent with recent experimental findings suggesting that individuals with ASD may update their priors slowly [11], putatively due to overestimating the volatility of the environment [10]. Further, here we suggest an explanation for the apparently inflexibility of priors in ASD. Namely, when resources are allocated primarily to represent statistically likely events, the representation of a given external environment may be best. However, it also limits the degree of resources that may be devoted to distinguishing between statistically unlikely alternatives [34,35]. These statistically unlikely events, however, are the most informative when attempting to infer whether the environment has changed. There is an inherent trade-off between coding for representational fidelity and for inference of changing environments given limited encoding resources [34,35]. Thus, the pattern of results suggesting reduced encoding resources in ASD may indicate that this deficit is at the root of aberrant perceptual flexibility in ASD [10,11]. This speculation is supported by a correlation between the total amount of encoding resources a participant demonstrated in the initial block of the experiment, and the degree to which their sensory encoding had adapted at the end of the experiment.

Related to the reallocation of sensory resources, studies on perceptual learning have identified a multitude of processes that may contribute to learning, even in simple tasks. These include changes in low-level representations, changes in attention, decision rules, and/or the assimilation of feedback [40,41]. Thus, one may wonder whether the observed inflexibility of sensory encoding in ASD reflects a more global learning deficit. We do not believe so, for the following reasons. First, the total amount of encoding resources differed between ASD and neurotypical individuals already at the outset of the experiment, and, as stated above, this quantity correlated with the degree of reallocation of encoding resources with feedback. Thus, if the inflexibility of encoding was expressing a global deficit in incorporating feedback, one would have to account for its correlation with total encoding resources at outset. Second, individuals with ASD did learn. Their allocation of encoding resources drifted from the natural toward the experimental distribution, particularly in wFB2 (**S6 Fig**), just not as readily as in neurotypical individuals. More importantly, their response times decreased over the course of the experiment, and this effect was in fact "larger" in ASD than neurotypical individuals (**S5 Fig**). Similarly, Harris and colleagues (2015) [42] have reported normal perceptual learning in individuals with ASD, in terms of improvement in discrimination threshold, as long as training and testing samples completely overlap (as was the case in the current experiment). It is only when probing for generalization in perceptual learning (i.e., from one retinotopic location to another) that poor performance is observed in ASD [42], again demonstrating a more specific, rather than a general, form of learning deficit.

There is a wide range of potential neural mechanisms by which the amount and allocation of encoding resources may change on a timescale that is compatible with the duration of our experiment. While neural recordings in animal models have shown that practicing orientation

discrimination can reshape tuning functions as early as V1 [43], given the timeframe of the current experiment, we consider that network-level mechanisms are most likely. A first possibility is a gain modulation of the neural response. Indeed, for an efficient coding framework where each cell transmits an equal portion of the stimulus probability mass, an increase in the overall firing rate corresponds to a direct increase in population FI [44]. Previous studies of sensory adaption (e.g., [26,45]) have also suggested that gain changes specific to a subpopulation of neurons are able to alter the allocation of coding resources. Gain modulation could be implemented by alterations in neuromodulation, and fittingly, Lawson and colleagues (2017) [10] recently demonstrated abnormal noradrenergic responsivity in ASD and suggested that this anomaly is related to the slow updating of their priors. The present findings suggest a functional interpretation for such a link: Abnormal noradrenergic responsivity could decrease FI (e.g., encoding resources), which then directly correlates with an inflexibility in altering priors. A second potential mechanism may be changes in the noise-correlation structure in the neural population. For example, a decrease in interneuronal correlations generally reduces the impact of noise on stimulus representation, thus leading to a higher population FI [46]. Along this line, Coen-Gagli and Solomon (2019) [47] have recently suggested that divisive normalization [48] is a critical player in neural variability, and Rosenberg and colleagues (2015) [49] have accounted for a wide array of perceptual deficits in ASD by postulating a deficit in divisive normalization.

For all these seemingly converging lines of evidence, however, there are open empirical questions.

First, in the current experiment, we provided participants with feedback on their performance in latter blocks of trials. This was done to accelerate learning of the novel statistical distribution of orientations and thus to enable reallocation of encoding resources. And indeed, scrutiny within the woFB block suggests that no learning occurred—neither in neurotypical nor ASD individuals—in the absence of feedback (**S6 Fig**), at least in the timeframe of our experiment. Thus, in the context of this experiment, seemingly feedback was necessary in reshaping encoding resources. This speculation goes hand in hand with our conjectures regarding the neural implementation of the current effect—likely being a network-level effect on the (feedforward) sensory representations in early visual areas (see [50] for an argument that the distribution of tuning functions in V1 underlies the oblique effect). In follow-up experiments, it will be interesting to repeat the current paradigm while considerably expanding the exposure time to the novel orientation distributions and without presenting feedback, in order to further understand if and how the updating of sensory encoding differs with mere exposure and with feedback.

Second, one may wonder about the potential impact of additional stimulus-independent variability introduced by factors following encoding—such as an inefficient decoder (i.e., a decoder that produces larger variance than that predicted by the Cramer–Rao lower bound), or post-perceptual factors such as motor noise. We attempted to experimentally minimize the potential impact of motor noise in ASD [51] by making the estimation task an unspeeded binary button press (i.e., up or down) that additionally had to be "confirmed" via another button press (see Materials and methods). Likewise, scrutiny of response time distributions did not suggest different strategies across experimental groups. Further, even if there were differences in post-encoding processes, these differing strategies (part of the decoding process) trade off biases and variances, and thus we would regardless recuperate an appropriate estimate of encoding (see **Fig 1** and **S1 Text**). Nevertheless, to further ascertain that potential stimuli-independent variability or post-perceptual factors do not deter from the current conclusions we ran additional simulations. We show that turning the decoder inefficient by adding homogeneous late noise will both decrease the total amount and flatten the pattern of extracted FI (**S7**

Fig). This is incompatible with the observed results in the following aspects. First, our empirical data suggest exposure to an artificial stimulus distribution coupled with feedback increases total FI (i.e., the λ parameter) while flattening the pattern of FI (i.e., the ω parameter). Thus, if the increase in total FI with feedback in the neurotypical group were to be explained by an increase in the efficiency in the decoder or a decrease in motor variability, the pattern of FI ought to be sharpened, which is the exact opposite of what we observed in the data. Similarly, if we were to explain the lack of change in total FI in the ASD group with increase in additional noise, the pattern of FI would have to go through even less learning (S8 Fig). Lastly, if we were to explain the initial difference in total FI (i.e., before feedback) between ASD and neurotypical individuals with additional stimulus-independent variability in the ASD group, then the "true" (i.e., uncorrupted by late noise) FI pattern in ASD would have to be more extreme (i.e., sharper at cardinal orientations) than what is specified by the stimulus distribution in the natural environment (see S8 Fig). While we cannot completely rule out this latter possibility, we consider that it is most parsimonious that both neurotypical and ASD groups do indeed start with a pattern of FI that is similarly adapted to the natural statistics of orientations, just as we observed in our experiment.

Third, here we have exclusively focused on sensory encoding in an effort to separately understand each piece leading to perception (i.e., encoding, recoding, decoding). However, comprehensively understanding sensory processing in ASD will ultimately necessitate a broader framework including encoding, prior beliefs, and loss (utility) functions, among others. In the context of oblique effect this will require a nontrivial form of encoding [23], orientation prior [21], and loss function [52]. Thus, instead of "reverse-engineering" all these components together [53,54], which would require a set of additional assumptions, here we took a data-driven approach. Our method, while not a complete model, can isolate and directly characterize the encoding process while being agnostic to potential differences in the decoding process and free of parametric restrictions about the shape of, for example, priors and likelihoods (see S1 Text for an extended discussion). As stated above, this is not to say that there are no differences in sensory decoding between the neurotypical and ASD group but to simply emphasize anomalies in encoding. Future work will be able to build on this foundation to extend the comparison of sensory processing in ASD relative to neurotypical individuals to a full model.

Lastly, we must mention that in a standard Bayesian framework, the lower encoding capacity of ASD would likely translate to broader likelihood functions or underweighting of sensory evidence. While some prior work [29,30,36,37] indeed suggest less precise sensory representation in ASD, others [55–58] have not replicated these observations. Butler and colleagues (2017) [58], for example, have shown no difference in the reliability of visual and somatosensory evoked activity during passive observation. Furthermore, some [3,54] have even argued for more, and not less, precise sensory likelihoods in ASD. In turn, future work will have to charter the domain generality of deficits and/or improvements in sensory encoding in ASD. More broadly, we suggest caution must be taken when comparing "likelihood functions" in an encoding–decoding framework across studies, given that these represent an abstract and aggregated property of a complex and multilayered hierarchical neural process, and could be strongly determined by factors such as stimulus type and task context. Here, therefore, instead of broadly describing potential anomalies in likelihood functions and/or prior distributions, we have taken a principled approach that links perceptual bias and variance to the available information in the encoding step. There are no degrees of freedom, and thus the problem is not underconstrained. Instead, FI is completely constrained by the measured pattern of biases and variance across the state space of interest (here 180 degrees of possible orientations). Further, by following the efficient coding hypothesis, we can relate encoding to perceptual priors

directly from psychophysical data without explicit parametric assumptions. Having taken these properties into account, the present analyses suggest that in ASD a reduced pool of encoding resources leads to increased variability, and then putatively to the inability to accurately detect and represent statistically unlikely events. This may explain the observed inflexibility in the reallocation of sensory resources following a change in the environment [10,11]. More importantly, our results highlight that the encoding stage itself appears aberrant in ASD, and thus future studies examining perception in ASD must carefully characterize other aspects of normative computation beyond Bayesian decoding.

## Materials and methods

All data and code for our analysis can be found at **S1 Code** or at https://github.com/cpc-lab-stocker/ASD_Encoding_2020

### Participants

A total of 42 participants completed an orientation-matching task. Seventeen were individuals diagnosed as within the ASD ($N = 17$, mean ± SD; age = 15.3 ± 2.6 years; AQ = 33.5 ± 7.0; SCQ = 16.8 ± 4.4) by expert clinicians. The rest were neurotypical individuals ($N = 25$, mean ± SD; age = 14.8 ± 2.1 years; AQ = 14.0 ± 5.5; SCQ = 5.6 ± 3.2). Participants had normal or corrected-to-normal vision and no history of musculoskeletal or neurological disorders. Before partaking in the study, all participants completed the Autism Spectrum Quotient (AQ; [59]) and the Social Communication Questionnaire (SCQ; [60]). The Institutional Review Board at Baylor College and Medicine approved this study (protocol number H-29411) in adherence to the Declaration of Helsinki, and all participants gave their written informed consent and/or assent.

### Materials and procedures

Participants were comfortably seated facing a gamma-corrected CRT monitor (Micron Technology, Boise, Idaho; 43 × 35 cm) at a distance of 57 cm. Participants self-initiated a trial by button press, upon which a Gabor (120 ms presentation, 0.4 cycles/degree, 10 degrees, Gaussian envelope) was presented centrally on a gray background. The orientation of this target Gabor was random (uniform distribution, 0 to 180 degrees). Immediately following the offset of the Gabor, a mask consistent of 6 concentric circles (line color: black; 2.5, 4, 5.5, 7, 8.5, and 10 degrees) was presented. This mask had a duration of 300 ms and was presented in order to eliminate the possibility of participants experiencing an afterimage. Following a blank period of 400 ms, participants were presented with a white Gabor patch (3 cycles/degree, only 1 strip visible, random initial orientation) that they rotated via button press (up and down arrow, resolution = 1 degree) until they considered the orientation of the white Gabor patch indicator to match that of the target Gabor. Participants logged a response via a "confirmatory" button press. The intertrial interval was set to 1 second, and participants completed 200 trials per block. The experiment consisted of 3 blocks; the first was without feedback, as described above. In the second and third blocks, participants were given feedback by superimposing the target Gabor and the orientation reported during the intertrial interval. Participants were given approximately 5-minute rest between blocks. All stimuli were generated and rendered using C++ Open Graphics Library (OpenGL).

### Data analysis

We model participants as a generic estimator $T$ of the orientation parameter $\theta$, and thus the responses as independent samples of $T$. The bias and variance of the estimator $T$ is a function

of $\theta$ defined by $b(\theta) = E_\theta[T] - \theta$, and $\sigma^2(\theta) = \sigma^2_\theta[T]$, respectively. A sliding window (of size $18°$ when analyzing individual participants and $4°$ for the combined participant) is used to calculate the function $b(\theta)$ and $\sigma^2(\theta)$. Within each window, due to the circular nature of orientation, the mean ($\mu$) and concentration ($\kappa$) parameters are calculated by fitting a von Mises distribution. The bias $b$ was then defined as the difference between $\mu$ and the center of the window. The variance $\sigma^2$ is defined as $\sigma^2 = 1/\kappa$, and standard deviation $\sigma = \sqrt{\sigma^2}$.

The lower bound of FI as a function of $\theta$ is given by Cramer–Rao lower bound (**Fig 1**):

$$\sqrt{I_F(\theta)} \geq \frac{[1 + b'(\theta)]}{\sigma(\theta)}. \tag{Eq 1, see main text}$$

For our analysis, we assumed a tight bound (e.g., a wide range of commonly used decoders, including Bayesian and maximum likelihood estimators can attain the bound, also see **Fig 1** and **S1 Text**), which allows us to extract FI directly from data. Furthermore, with an efficient coding assumption that the square-root of FI is directly proportional to the prior distribution of the stimulus:

$$\sqrt{I_F(\theta)} \sim p(\theta). \tag{Eq 2}$$

We can thus estimate the prior distribution that corresponds to an efficient sensory representation directly from data by calculating the normalized square-root of FI (**Fig 4A**):

$$\sqrt{I_F(\theta)} / \int_\theta \sqrt{I_F(\theta)} \, d\theta. \tag{Eq 3}$$

Note that the denominator, $\int_\theta \sqrt{I_F(\theta)} \, d\theta$, is also a direct measure of the total amount of resource available for encoding (**Fig 4B**; [16,23]).

Next, to quantify the changes in encoding during the experiment, we parameterized the square-root of FI with $\lambda$ and $\omega$:

$$\sqrt{I_F}(\theta; \lambda, \omega) = \lambda * \left( \omega * [0.877 * (1 - sin|2\theta|)] + (1 - \omega) * \frac{1}{\pi} \right), \tag{Eq 4}$$

where $\lambda$ determines the amount of total (square-root of) FI, and $\omega$ controls the shape of FI (i.e., allocation of encoding resources, effectively the prior distribution under the efficient coding hypothesis) by mixing a "cardinal orientation prior" with a uniform orientation distribution (**Fig 4C**). Note the constants are such that the prior term of the equation normalizes properly to 1.

To recover $\lambda$ and $\omega$, we calculate the predicted bias:

$$\hat{b}(\theta; \lambda, \omega) = \int_0^\theta (\sqrt{I_F}(\theta; \lambda, \omega) * \sigma(\theta) - 1) d\theta. \tag{Eq 5}$$

We find $\hat{\lambda}$ and $\hat{\omega}$ that give rise to the best fit to the observed bias $b(\theta)$ using MATLAB's *fmincon*:

$$argmin_{\lambda,\omega} ||\hat{b}(\theta; \lambda, \omega) - b(\theta)||_2^2. \tag{Eq 6}$$

See **S1** and **S2 Figs** for individual participant fits. See **S3** and **S4 Figs** for parameters extracted for each individual participant and their relationship with measures of autism symptomatology. Note that in principle, we can fit $\sqrt{I_F}(\theta; \lambda, \omega)$ to the extracted FI directly. Here,

we choose to fit the bias pattern $b(\theta)$, to avoid the noisy derivative, $b'(\theta)$ in the Cramér–Rao lower bound, and effectively replaced it with an integration step instead.

Fig 4D and 4E show the parameters estimated for the combined participant, while the regression analysis presented in Fig 5 is based on parameters estimated with the same procedure applied to each individual participant. All statistical tests and $p$-values (except for the regression in Fig 5) are two-sided and are based on the distribution (intervals) of the sample statistics (e.g., mean, variance, model parameters $\omega$ and $\lambda$) across 5,000 bootstrap runs ([61]).

## Supporting information

**S1 Text. Motivation and analytical derivation for using the Cramer–Rao lower bound to measure Fisher information of the encoding process.**
(DOCX)

**S1 Fig.** Scatter plot of the response data for individual neurotypical participants (target on x-axis and bias on y-axis), and the model fits (dotted lines) to the average bias (solid lines). The number on the top right of each panel indicates the goodness-of-fit ($R^2$). Average $R^2$: 0.770 ±0.025. Note that we enforced the bias pattern to have a periodicity of 90˚. Color gradient (from dark to light) and left-to-right shows first the block without feedback (woFB), and then the first and second blocks with feedback (wFB1, wFB2). Raw data and code underlying this figure can be found at S1 Code, and numerical values that make up this figure can be found at S6 Data.
(EPS)

**S2 Fig.** Scatter plot of the response data for individual ASD participants (target on x-axis and bias on y-axis), and the model fits (dotted line) to the average bias (solid line). The number on the top right of each panel indicates the goodness-of-fit ($R^2$). Average $R^2$: 0.774±0.034. Note that we enforced the bias pattern to have a periodicity of 90˚. Color gradient (from dark to light) and left-to-right shows first the block without feedback (woFB), and then the first and second blocks with feedback (wFB1, wFB2). Raw data and code underlying this figure can be found at S1 Code, and numerical values that make up this figure can be found at S6 Data. ASD, autism spectrum disorder.
(EPS)

**S3 Fig.** Magnitude $\lambda$ (top) and shape $\omega$ (bottom) of FI as a function of AQ scores for individual participants. Neurotypical individuals are depicted in blue, and ASD participants are shown in red (individual dots are single participants, dots with errors bars are means across individuals ± SEM). The results corroborate the conclusion (Fig 4D and 4E) that the neurotypical and ASD group differed only by the $\lambda$ parameter before feedback but both the $\lambda$ and $\omega$ parameters after feedback. Raw data and code underlying this figure can be found at S1 Code, and numerical values that make up this figure can be found at S5 Data. AQ, Autism Quotient; ASD, autism spectrum disorder; FI, Fisher information.
(EPS)

**S4 Fig.** Magnitude $\lambda$ (top) and shape $\omega$ (bottom) of FI as a function of SCQ scores. Raw data and code underlying this figure can be found at S1 Code, and numerical values that make up this figure can be found at S5 Data. FI, Fisher information; SCQ, Social Communication Questionnaire.
(EPS)

**S5 Fig. Histogram of RT (i.e., time it took for participants to rotate the response Gabor patch to match the target orientation).** The overall shape of the distributions is qualitatively

similar. There is a significant albeit small difference in the median RT between the neurotypical and ASD group across all 3 blocks (woFB, wFB1, and wFB2). Furthermore, there is a significant decrease in RT for both the neurotypical and ASD group with feedback as the experiment progresses. Raw data and code underlying this figure can be found at **S1 Code**, and numerical values that make up this figure can be found at **S6 Data**. ASD, autism spectrum disorder; RT, response time.
(EPS)

**S6 Fig.** We conducted a sliding-window analysis in an attempt to reveal the effect of learning within each block (woFB, wFB1, and wFB2) in terms of changes in: (**A**) average magnitude of the bias; (**B**) standard deviation of the response; (**C**) shape of the FI $\omega$; and (**D**) magnitude of the FI $\lambda$. Our analysis suggested that, within the context of our experiment, feedback seems essential for learning. In fact, within the first block (woFB), the bias and variance of both groups seem to be increasing as opposed to decreasing. Raw data and code underlying this figure can be found at **S1 Code**, and numerical values that make up this figure can be found at **S6 Data**. FI, Fisher information.
(EPS)

**S7 Fig.** (**A**) We simulated the potential impact of stimulus-independent noise/variability introduced by factors other than encoding, such as an inefficient decoder (i.e., a decoder that produces larger variance than that dictated by the Cramer–Rao lower bound), or post-decisional factors such as motor noise. We consider this important, given that while we attempted to reduce motor responses to their simplest form (i.e., nonspeeded binary button press as opposed to more complex movements such as joystick use), the presence of increase motor noise is well documented in ASD (e.g., Gowen and Hamilton, 2012). (**B**) An increase in stimulus-independent, post-encoding noise will cause a decrease in the overall magnitude of the extracted FI. (**C**) An increase in post-encoding noise will flatten the shape of the extracted FI, while decrease the overall magnitude of FI. These results are incompatible with what we observed in our experiment, since we observe either an increase in the magnitude of FI associated with a flattening in the shape of FI (neurotypical group), or no change in the magnitude of FI, while the shape of FI is slightly flattened (ASD). Therefore, these simulations argue that the differences we herein report cannot be attributed to motor noise or other sources of noise that are stimuli-independent (see Discussion and **S8 Fig**). Raw data and code underlying this figure can be found at **S1 Code**, and numerical values that make up this figure can be found at **S6 Data**. ASD, autism spectrum disorder; FI, Fisher information.
(EPS)

**S8 Fig. In a further attempt to understand the potential impact of post-encoding noise, we conducted model simulations with the counterfactual assumption that the abnormalities in $\lambda$ (both total and change thereof) in the ASD group are fully explained by stimulus-independent noise.** (**A**) We simulated an artificial observer (black solid) for which its FI increased during wFB2, just as for neurotypical individuals. However, due to additional noise, this observer appears (black dashed) to be unchanged, similar as the ASD group (red dashed). (**B**) The $\omega$ values in the artificial observer (black solid) would have to undergo less learning than what was observed in ASD, such that that when it is corrupted by additional noise, it appears to have similar $\omega$ values as the ASD group (red dashed). (**C**) Similarly, we simulated another artificial observer (black solid) for which its FI matched that of the neurotypical individuals during woFB (blue dashed), yet appears to be reduced (black dashed) to a similar level as the ASD group (red dashed) due to additional noise. (**D**) The $\omega$ values in the artificial observer (black solid) would have to be even more extreme (FI peaking even stronger than what is

determined by the natural distribution of orientations in the environment), such that when it is corrupted by additional noise, it appears to have similar $\omega$ values as the ASD group (red dashed). We consider this unlikely since it is most parsimonious that both neurotypical and ASD groups do indeed start with a pattern of FI that is similarly adapted to the natural statistics of orientations. Importantly, in both cases, the $\omega$ values in the simulated observer strongly support our main conclusion that the neurotypical group reallocate their encoding resources toward uniform distribution more so than the ASD group did. Raw data and code underlying this figure can be found at **S1 Code**, and numerical values that make up this figure can be found at **S6 Data**. ASD, autism spectrum disorder; FI, Fisher information.
(EPS)

**S1 Code. Raw data and MATLAB code for our analysis and figures.** Routine to reproduce Figs 1D–1F, 2B, 2C, 3, 4 and 5 and S1–S6, panel B in S7, and S8 Figs.
(ZIP)

**S1 Data. Numerical values that make up Fig 1D–1F.**
(ZIP)

**S2 Data. Numerical values that make up Fig 2B and 2C.**
(ZIP)

**S3 Data. Numerical values that make up Fig 3.**
(ZIP)

**S4 Data. Numerical values that make up Fig 4.**
(ZIP)

**S5 Data. Numerical values that make up Fig 5 and S3 and S4 Figs.**
(ZIP)

**S6 Data. Numerical values that make up S1, S2, S5, S6, panel B in S7, and S8 Figs.**
(ZIP)

## Acknowledgments

We thank J. Patterson and A. Rosenberg for piloting some of the stimuli used in the present experiments and H. Park for participation in data collection.

## Author Contributions

**Data curation:** Jean-Paul Noel, Ling-Qi Zhang.

**Formal analysis:** Jean-Paul Noel, Ling-Qi Zhang.

**Funding acquisition:** Dora E. Angelaki.

**Investigation:** Jean-Paul Noel, Ling-Qi Zhang, Alan A. Stocker.

**Methodology:** Jean-Paul Noel.

**Project administration:** Alan A. Stocker, Dora E. Angelaki.

**Software:** Ling-Qi Zhang.

**Supervision:** Alan A. Stocker.

**Validation:** Alan A. Stocker.

**Visualization:** Jean-Paul Noel, Ling-Qi Zhang.

**Writing – original draft:** Jean-Paul Noel, Ling-Qi Zhang, Alan A. Stocker, Dora E. Angelaki.

**Writing – review & editing:** Jean-Paul Noel, Ling-Qi Zhang, Alan A. Stocker, Dora E. Angelaki.

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
