## [Editor Report · Decision Letter 0]

15 Sep 2020

Dear Dr Noel, 

Thank you for submitting your manuscript entitled "Aberrant Sensory Encoding in Patients with Autism" for consideration as a Research Article by PLOS Biology.

Your manuscript has now been evaluated by the PLOS Biology editorial staff as well as by an academic editor with relevant expertise and I am writing to let you know that we would like to send your submission out for external peer review.

Please re-submit your manuscript within two working days, i.e. by Sep 17 2020 11:59PM.

Kind regards,

Lucas Smith, Ph.D.,

Associate Editor

PLOS Biology

---

## [Decision Letter · Decision Letter 1]

12 Nov 2020

Dear Dr Noel,

Thank you very much for submitting your manuscript "Aberrant Sensory Encoding in Patients with Autism" for consideration as a Research Article at PLOS Biology. Your manuscript has been evaluated by the PLOS Biology editors, an Academic Editor with relevant expertise, and by several independent reviewers.

The reviews of your manuscript are appended below. As you will see from their comments, all of the reviewers felt that the topic of this study was interesting, however the reviewers have raised potential issues with the experimental paradigm, and feel that in places the conclusions of the study are not fully supported by the data.

In light of the reviews, we will not be able to accept the current version of the manuscript, but we would welcome re-submission of a much-revised version that takes into account the reviewers' comments and addresses the specific points raised by the reviewers. Please pay special attention to the concerns raised by Reviewer 1, which should be thoroughly and satisfactorily addressed.

We expect to receive your revised manuscript within 3 months. 

**IMPORTANT - SUBMITTING YOUR REVISION**

*Re-submission Checklist*

*Published Peer Review*

*PLOS Data Policy*

*Blot and Gel Data Policy*

Sincerely,

Lucas Smith, Ph.D.,

Associate Editor,

lsmith@plos.org,

PLOS Biology

REVIEWS:

Reviewer's Responses to Questions

PLOS authors have the option to publish the peer review history of their article (what does this mean?). If published, this will include your full peer review and any attached files.

Reviewer #1: No

Reviewer #2: Yes: Albert Powers

Reviewer #3: Yes: Duje Tadin

Reviewer #1: The authors use the oblique effect to examine sensory encoding in neurotypical and autistic individuals. They report that after undergoing trials with feedback, controls, but not autistic participants change their sensory representations and 'increase encoding capacity'. This, they conclude, suggests deficits in sensory encoding in autism and compromised ability to adapt to novel input statistics. I find the notion of autism being associated with inflexible priors quite appealing, given its potential explanatory scope for many facets of the phenotype. However, I am not persuaded that the empirical evidence presented here provides unequivocal support for this proposal. My concerns are related to the experimental paradigm used and also the interpretation of some of the findings.

The oblique effect strikes me as a less than ideal phenomenon to use for probing dynamic changes in priors. Although the neural basis of this effect is not definitively established, studies that have examined responses from large collections of neurons in visual cortex suggest that anisotropies in cell preferences as early as V1 may largely be responsible for the phenomenon (e.g. Baowang Li, Matthew R. Peterson, and Ralph D. Freeman (2003). Oblique Effect: A Neural Basis in the Visual Cortex. J Neurophysiol 90: 204-217). Response properties of neurons in early sensory cortex, that have been shaped by long evolutionary history, are not readily modifiable by short experiences. Hence, the oblique effect may not be the ideal vehicle to assess the impact of short-term changes in image statistics on sensory encoding.

The task that participants are asked to perform relies not just upon encoding, but also short-term memory (to remember the seen orientation until a response is made) and the ability to avoid 'contamination' of the remembered orientation by the perceived orientation of the response Gabor itself. In this context, can the differences in performance between controls and ASD participants be unambiguously ascribed to encoding differences alone? It is possible that ASD participants are more susceptible to contamination between response probe orientation and remembered orientation, and hence do not show the kinds of gains in performance that controls do across the experimental blocks. (I might have missed this in the manuscript, but it is important to state how long it takes subjects in the two groups to respond, i.e. reorient the probe Gabor from its random starting orientation to the final one. How monotonic is this process? Do subjects exhibit overshoots and corrections? Do the different groups differ in this? Might ASD participants be more or less impulsive in their responses relative to controls and might this impact their response bias? All of this is to make the point that the process of responding may be somewhat protracted and involve inadvertent modifications of the remembered target orientation.)

I am also skeptical of the suggestion that controls' priors are changed by exposure to stimuli with a uniform distribution of orientation. If that were to be the case, exposure alone should have sufficed; feedback should not have been necessary. Since anisotropies in orientation encoding are believed to arise from mere exposure to a world with increased representation of cardinal directions, why should the modification of these priors require feedback? 

On a minor note, the classic oblique effect is associated with high orientation discrimination accuracy for horizontal and vertical orientations ('cardinal' directions) and least for obliques. In the data reported here, cardinal as well as oblique directions elicit high accuracies. What might account for this discrepancy? Might it be an outcome of the memory component of the task used here, with the four orientations being recalled symbolically ('horizontal', 'vertical', 'left diagonal', 'right diagonal')?

Overall, while I am sympathetic to the conceptual point the manuscript is attempting to make, I am not entirely convinced that the empirical data are adequately clear in the interpretations they allow.

Reviewer #2: In this ambitious paper, the authors attempt to take advantage of the so-called "oblique effect" to estimate aberrant sensory encoding in ASD. This is a really interesting and clever approach, but I have several concerns that significantly impact my enthusiasm for the manuscript in its current form. These are enumerated below. 

1. Authors make the case that encoding and decoding can be isolated behaviorally. However, the fullness of pathophysiology may come from the pairing between aberrant sensory encoding and some decoding scheme.

2. Relatedly, others have demonstrated high-precision likelihoods in ASD. Why would this arise in the context of low-fidelity sensory encoding? Wouldn't one expect the opposite? Or is it the authors' contention that these two deficits are unrelated? This should be explained more fully in the Discussion. 

3. Bias and variance (Fig 1), observed behaviorally, are taken to be direct read-outs of the percept resulting from decoding. But of course this cannot be—a response is generated somehow and a response model must be specified if behavioral responses are used to estimate the quantities included in the model. What is this response model? If there is none,why not? 

4. The authors would do well to motivate the use of feedback blocks during the introduction. Feedback comes somewhat as a surprise when introduced in the Results section and

the reader is unclear why this manipulation was introduced in the context of the overall study.

5. Related to 3, and especially in the context of feedback-based learning, it is not surprising that he authors how an improvement in FI while showing a decrease in bias. There are no other variables capable of explaining that difference, given the authors' model. This doesn't mean that the results are not due to simple response bias or any other post-encoding step of processing, but rather that the model, in its incompleteness, is incapable of accounting for any other reason for the behavioral results. The authors provide reasons why behavioral noise and decoding processes are unlikely to account for the results observed. But the only way to make this truly convincing is to test it formally by including a Bayesian decoding and response model with relevant parameters included.

Reviewer 3: This study investigates sensory encoding in autism (ASD). Over the past decade, there has been an increase in work looking at sensory abnormalities in autism, but we still lack a good understanding of how sensory, namely visual, processing is atypical in autism. This study focuses on visual encoding, a critical early step in visual processing. This question is addressed in the context of visual orientation discrimination and combines convincing behavioral data (I’d say remarkably convincing given what is typically seen in ASD work) and sophisticated computational modeling. The conclusion is that sensory encoding is severely impaired in ASD.

I find this to be an interesting and a noteworthy study. The behavioral data are, to repeat, convincing and certainly indicate atypical orientation processing in ASD. The modeling is well done, and while I’m not ready to fully “buy” the ultimate conclusions (details below), even if those conclusions turn out to be not entirely correct, this work will still be an important addition to our understanding of how sensory processing could be impaired in ASD.

1A. There are two key behavioral findings: (1) the overall difference in performance in the first block and (2) poor learning/adaptation of the ASD group over blocks. Both findings need to be better integrated with the existing literature, both to better contextualize the present results and to consider alternative explanations. The authors do link the first result to other studies showing increased variability in ASD, but more is needed there. For example, a recent study that examined variability of visual orientation processing is not mentioned. Also, it is worth mentioning work that questions the “noise hypothesis” in ASD. Both references are below.

Park, W. J., Schauder, K. B., Zhang, R., Bennetto, L., & Tadin, D. (2017). High internal noise and poor external noise filtering characterize perception in autism spectrum disorder. Scientific reports, 7(1), 1-12.

Butler, J. S., Molholm, S., Andrade, G. N., & Foxe, J. J. (2017). An examination of the neural unreliability thesis of autism. Cerebral cortex, 27(1), 185-200.

1B. In addition, the change in performance over blocks is discussed as learning, but no work on perceptual learning in ASD is mentioned. For example, the paper listed below showed that perceptual learning in ASD is atypical. This raises the question of whether the present results could be a result of a boarder deficit in learning?

Harris, H., Israeli, D., Minshew, N., Bonneh, Y., Heeger, D. J., Behrmann, M., & Sagi, D. (2015). Perceptual learning in autism: over-specificity and possible remedies. Nature neuroscience, 18(11), 1574-1576.

2. The first block does not use feedback, while the 2nd and the 3rd blocks do. That decision needs to be explained and justified, but it is not. Why was adding feedback after the first block important? After all, feedback is not necessary for us to learn stimulus statistics (though it likely speeds up the process).

My worry is that this leaves open a possibility that how the two groups responded to this change in feedback

could be, in part, what drives the difference. For example, what if individuals with ASD ignored feedback? (btw, this is a hypothetical, I don’t have a reason to believe this). What I’m looking for here is a better justification of why the feedback was used in this specific way and are there good reasons to believe that the results would generalize to other ways of deploying feedback (or no feedback at all). That is, I’m not asking for new data (certainly not during COVID times).

3. The AQ results (Fig 5) are entirely un-informative. The reported correlations seem to be mostly driven by the overall group differences, with little evidence of within-group correlation effects. For these conclusions to stand, what is needed is evidence that **within** the ASD group, these results are corelates with the symptom severity.

Moreover, AQ is a somewhat outdated measure that relies a lot on imperfect stereotypes of what autism is, and it is not used in the diagnosis process (rather it is best used as a measure of autism-like behaviors and thinking among the general population). However, it does look like ADOS was collected.

Minor points:

-- I’d take out “patients” from the title. Many individuals with autism do not consider themselves to be patients.

-- In discussion (page 9), it is stated “This speculation will have to be confirmed in physiological studies.” It is not clear how. Are we talking about EEG and fMRI or animal models?

-- Materials and Procedures: stimulus sizes need to be given in visual degrees and not centimeters.

---

## [Decision Letter · Decision Letter 2]

23 Mar 2021

Dear Dr Noel,

Thank you for submitting your revised Research Article entitled "Aberrant Visual Orientation Encoding in Individuals with Autism" for publication in PLOS Biology. I have now obtained advice from the original reviewers and have discussed their comments with the Academic Editor. 

The reviewers are all satisfied with the revised manuscript and think their previous concerns have been satisfactorily addressed. Therefore we will probably accept this manuscript for publication, provided you address the remaining editorial, data, and other policy-related requests included here and in more detail below my signature.

1) Thank you for adapting your title in response to reviewer 3's comment that "Many individuals with autism do not consider themselves to be patients". Along these lines, we think the rest of the manuscript should be edited to reflect this. We think language calling individuals with ASD 'patients' should be changed throughout the manuscript and abstract. Similarly, we think that instead of calling the control group 'healthy' you might use 'neurotypical'. 

2) We would also like to suggest an edit to the title, which we think might help make it a bit more direct and clear. If you agree, we think the title could be changed to "People with autism spectrum disorder have altered visual encoding capacity".

3) Ethics request: Please indicate whether your protocol, approved by the Baylor College and Medicine IRB adhered to the Declaration of Helsinki or other national or international ethical guidelines.

4) Ethics Request: Please include the ID number of the protocol approved by the Baylor College and Medicine IRB.

5) Data Request: Please provide, as a supplementary file, the summary statistics used to generate each graph presented in your study. Please make sure to provide a legend for this file, and to refer to it in your data availability statement and figure legends (including supplementary figure legends). For example, to each figure legend, you could add a statement saying “The data underlying this figure can be found in the supplementary file S1_data”.

We expect to receive your revised manuscript within two weeks. 

*Published Peer Review History*

*Early Version*

Sincerely,

Lucas Smith, Ph.D.,

Associate Editor,

lsmith@plos.org,

PLOS Biology

ETHICS STATEMENT:

-- Please indicate whether your protocol, approved by the Baylor College and Medicine IRB adhered to the Declaration of Helsinki or other national or international ethical guidelines

-- Please include the ID number of the protocol approved by the Baylor College and Medicine IRB.

DATA POLICY:

Fig 1d-f; Fig 2B-C; Fig 3A-F; Fig 4 A-B,D-E; Fig 5A-B; Fig S1; Fig S2; Fig S3; Fig S4; Fig S5; Fig S6; Fig S7B; Fig S8A-D

Reviewer remarks:

Reviewer #1: The authors have thoughtfully responded to the concerns I had raised in my earlier review. I believe the manuscript is now suitable for publication.

Reviewer #2: The authors have done an impressive job of responding to my comments comprehensively. I have no further comments.

Reviewer #3 (Duje Tadin): I appreciate the authors' careful work on this revision. It addresses the concerns I had about this manuscript, namely concerns about links with other literature on sensory noise and on learning. I also appreciate the detailed consideration and additional analyses on the role of feedback, and how the authors responded to concerns about the AQ analysis.

---

## [Editor Report · Decision Letter 3]

1 Apr 2021

Dear Dr Noel,

Thank you very much for addressing our last editorial requests in a revised manuscript. On behalf of my colleagues and the Academic Editor, Adam Kohn, I am pleased to say that we can now, in principle offer to publish your Research Article "Individuals with Autism Spectrum Disorder Have Altered Visual Encoding Capacity" in PLOS Biology, provided you address any remaining formatting and reporting issues. These will be detailed in an email that will follow this letter and that you will usually receive within 2-3 business days, during which time no action is required from you. Please note that we will not be able to formally accept your manuscript and schedule it for publication until you have made the required changes.

When addressing these last formatting requests, we also ask that adjust your data availability statement to reference the recently added supplementary files. For example, you could say “The data underlying each figure can be found in supplementary files. Additional data and code for our analysis can be found at: https://github.com/cpc-lab-stocker/ASD_Encoding_2020”

PRESS

Thank you again for supporting Open Access publishing. We look forward to publishing your paper in PLOS Biology. 

Sincerely, 

Lucas Smith, Ph.D. 

Associate Editor 

PLOS Biology